# Reduced serotonergic transmission alters sensitivity to cost and reward via 5-HT$_{1A}$ and 5-HT$_{1B}$ receptors in monkeys

Yukiko Hori[1], Koki Mimura[1,2], Yuji Nagai[1], Yuki Hori[1], Katsushi Kumata[3], Ming-Rong Zhang[3], Tetsuya Suhara[1], Makoto Higuchi[1], Takafumi Minamimoto[1]*

1 Department of Functional Brain Imaging, National Institutes for Quantum Science and Technology, Chiba, Japan, 2 Research Center for Medical and Health Data Science, The Institute of Statistical Mathematics, Tokyo, Japan, 3 Department of Advanced Nuclear Medicine Sciences, National Institutes for Quantum Science and Technology, Chiba, Japan

* minamimoto.takafumi@qst.go.jp

**Data Availability Statement:** The data underling all the figures will be available on the following public repository: https://doi.org/10.5281/zenodo.10141750.

## Abstract

Serotonin (5-HT) deficiency is a core biological pathology underlying depression and other psychiatric disorders whose key symptoms include decreased motivation. However, the exact role of 5-HT in motivation remains controversial and elusive. Here, we pharmacologically manipulated the 5-HT system in macaque monkeys and quantified the effects on motivation for goal-directed actions in terms of incentives and costs. Reversible inhibition of 5-HT synthesis increased errors and reaction times on goal-directed tasks, indicating reduced motivation. Analysis found incentive-dependent and cost-dependent components of this reduction. To identify the receptor subtypes that mediate cost and incentive, we systemically administered antagonists specific to 4 major 5-HT receptor subtypes: 5-HT$_{1A}$, 5-HT$_{1B}$, 5-HT$_{2A}$, and 5-HT$_4$. Positron emission tomography (PET) visualized the unique distribution of each subtype in limbic brain regions and determined the systemic dosage for antagonists that would achieve approximately 30% occupancy. Only blockade of 5-HT$_{1A}$ decreased motivation through changes in both expected cost and incentive; sensitivity to future workload and time delay to reward increased (cost) and reward value decreased (incentive). Blocking the 5-HT$_{1B}$ receptor also reduced motivation through decreased incentive, although it did not affect expected cost. These results suggest that 5-HT deficiency disrupts 2 processes, the subjective valuation of costs and rewards, via 5-HT$_{1A}$ and 5-HT$_{1B}$ receptors, thus leading to reduced motivation.

## Introduction

The motivation to do any voluntary goal-directed behavior is a trade-off between incentive and the expected cost. The subjective value of the reward expected upon achieving the goal determines the incentive (or incentive value), which has a positive influence on motivation. The expected cost is a subjective valuation of what must be paid to earn the reward (e.g., waiting, taking a risk, or putting forth effort) and has a negative influence [1–3]. Thus, motivation

**Funding:** This work was supported in part by KAKENHI Grants JP22K07339 (to YH), and JP26120733, JP18H04037 and JP20H05955 (to TM) from Japan Society for the Promotion of Science (JSPS), and by the Moonshot Research & Development Program (Millennia Program) from Japan Science and Technology Agency (JST) Grant Number JPMJMS2295 (to TM). The funders had no role in study design, data collection and analysis, decision to publish, or preparation of the manuscript.

**Competing interests:** The authors have declared that no competing interests exist.

**Abbreviations:** 5-HIAA, 5-hydroxyindoleacetic acid; BIC, Bayesian information criterion; CU, cost unit; CSF, cerebrospinal fluid; DA, dopamine; HSD, honest significant difference; ITI, inter-trial interval; LMM, linear mixed model; mPFC, medial prefrontal cortex; MR, magnetic resonance; pCPA, para-chlorophenylalanine; PET, positron emission tomography; PTSD, post-traumatic stress disorder; RT, reaction time; VOI, volume of interest.

will be highest when high-value rewards can be obtained at low cost and vice versa. Deficiencies in the serotonin (5-HT) system are known to disturb incentive and expected cost, which in turn leads to abnormal motivation. Indeed, depression and other psychiatric disorders are associated with 5-HT deficiency, as 5-HT has become the target of numerous medications for treating these conditions. 5-HT deficiency is currently thought to reduce incentive (e.g., blunting/eliminating pleasure from rewards) and/or increase the expected cost (e.g., waiting becomes intolerable, resulting in impulsivity) [4,5]. In rodent studies, both 5-HT depletion and attenuation of 5-HT transmission via specific receptor antagonists increases impulsive choices [6–11] and reduces the frequency of effortful behavior [12]. In addition, pharmacological attenuation of dorsal raphe 5-HT neurons has been shown to impair the ability/desire to wait for long-delayed rewards [13]. Yet, despite this evidence, the specific mechanisms through which 5-HT contributes to motivation remain unclear, especially in primates.

Understanding the role of 5-HT in motivation is challenging, in part because incentives and costs have not been adequately considered in the 5-HT literature. Indeed, the behavioral effects of reducing 5-HT activity in the brain have never been quantitatively and independently examined in terms of incentives and costs. The fourteen 5-HT receptor subtypes in the central nervous system [14] and their heterogeneous localizations are another factor that complicates the issue. Therefore, characterizing the role of each receptor in modulating motivation via incentive and/or cost is critical, as is determining their locations in the brain.

In a previous study, we demonstrated dissociable roles of the 2 dopamine (DA) receptor subtypes in computing the cost-benefit trade-off in motivation. This was accomplished by combining positron emission tomography (PET), pharmacological manipulation of DA receptors, and quantitative measures of motivation in monkeys [15]. Here, we used the same methodology to examine the 5-HT system and determine how 5-HT neurotransmission affects the motivation for goal-directed behavior. First, we manipulated the 5-HT system by repeatedly administering para-chlorophenylalanine (pCPA), a reversible inhibitor of 5-HT synthesis, to macaque monkeys and examining its effects on goal-directed behavior. We then focused on four 5-HT receptor subtypes (5-HTRs; specifically, $5\text{-HT}_{1A}R$, $5\text{-HT}_{1B}R$, $5\text{-HT}_{2A}R$, and $5\text{-HT}_4R$) that are abundant in limbic (motivation-related) brain regions [16]. We mapped their distributions and manipulated 5-HT transmission by systemically administering receptor type-specific antagonists at doses predetermined by PET to achieve the same degree of receptor occupancy. Behavioral effects were assessed using 2 tasks designed to examine incentive and expected cost, respectively. Our results suggest that a reduction in 5-HT transmission leads to reduced motivation through 2 distinct processes: increased cost sensitivity (expected cost) via $5\text{-HT}_{1A}R$ and reduced reward impact (incentive) via $5\text{-HT}_{1A}R$ and $5\text{-HT}_{1B}R$.

## Results

### Effects of 5-HT depletion on incentive

We first determined how much 5-HT is depleted by preventing its synthesis with pCPA. We repeatedly injected pCPA (150 mg/kg, s.c.) over the course of 2 days, which resulted in a decrease in 5-HT metabolites (5-hydroxyindoleacetic acid, 5-HIAA) of at least 30% ($N = 2$, 33% and 64%) in the cerebrospinal fluid (CSF), while the concentration of DA remained unchanged (Fig 1A and 1B). The 30% value was later used when determining the dosages for specific 5-HT receptor-type antagonists.

We next examined the effects of 5-HT depletion on incentive in 4 monkeys that were not used in the CSF study (S1 and S2 Tables). For this purpose, we used a reward-size task in which the amount of reward was manipulated across trials, but the task requirements (i.e., the costs) remained the same (Fig 1C). For each trial, the monkeys could receive a reward if they

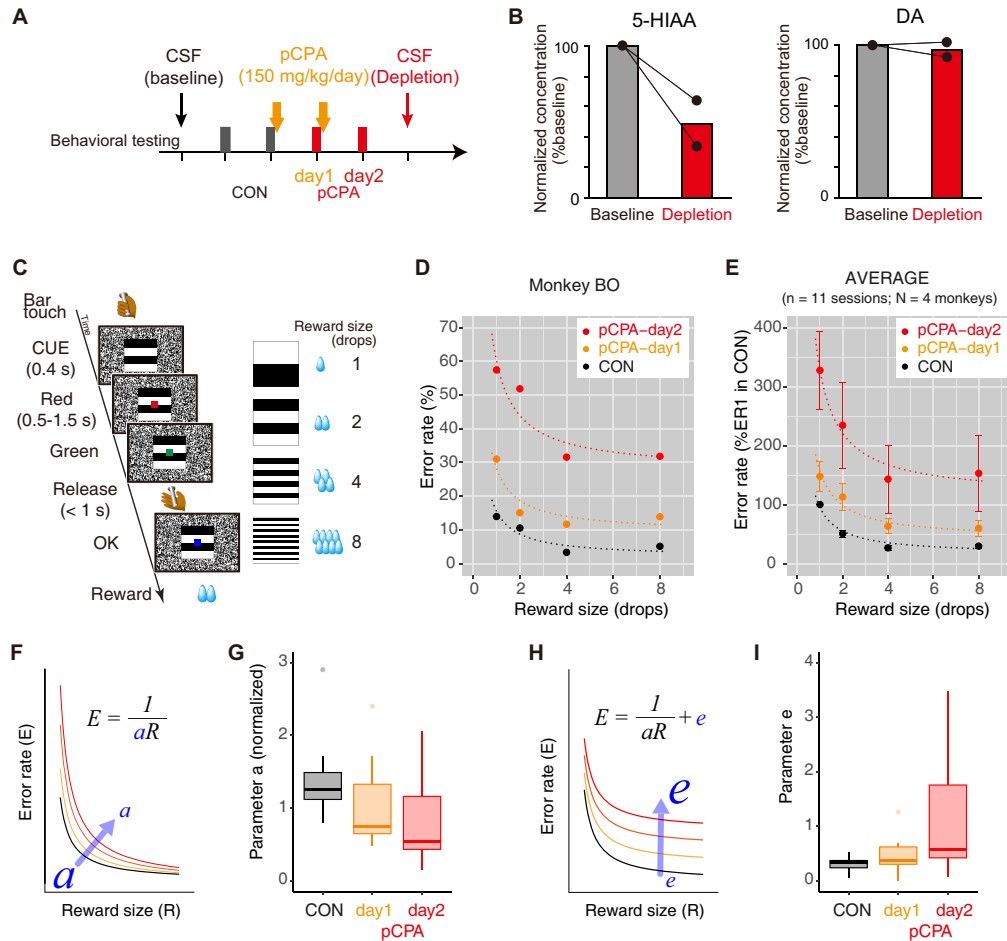

**Fig 1. Effect of 5-HT depletion via pCPA on incentive.** (A) Schedule of pCPA treatment, behavioral testing, and CSF sampling. (B) Normalized concentration of 5-HT metabolite (5-HIAA) and DA before (baseline) and after pCPA treatment (depletion). (C) Reward-size task. Left: Sequence of events during 1 trial. A monkey initiated a trial by touching the bar in the chair. After 100 ms, a visual cue signaling the amount of reward (1, 2, 4, or 8 drops) that would be delivered was presented at the center of the monitor. After 500 ms, a red target also appeared at the center of the monitor. After a variable interval of 500–1,500 ms, the target turned green, indicating that the monkey could release the bar to receive the reward. If the monkey responded between 200 and 1,000 ms, the target turned blue indicating the trial had been completed correctly. On correct trials, water rewards were delivered immediately. An ITI of 1 s was enforced before the next trial could begin. If the monkey made an error by releasing the bar before the green target appeared, within 200 ms after it appeared, or failed to respond within 1 s, all visual stimuli disappeared, the trial was terminated immediately, and the trial was repeated after the 1-s ITI. After each correct trial, a new cue reward-size pair was picked from the set of 4 at random. Right: Relationship between visual cues and reward size. (D) Representative error rates as a function of reward size for monkey BO. Dotted curves are the best fit of inverse functions (Model #4, S3 Table). (E) Normalized error rate (percent of maximum error rate in 1 drop trial in the control session; mean ± SEM) as a function of reward size for $n = 11$ sessions collected from 4 monkeys (S2 Table). (F) Schematic illustration of increase in error rate as incentive $a$ is reduced. (G) Box plot of normalized incentive ($a$) for each treatment condition ($n = 11$ for each). Each value was normalized to the value of the control condition. (H) Schematic explaining the increase in error rate by $e$, independent of reward size. (I) Box plot of parameter $e$ (normalized as the ratio of maximum error rate in 1 drop trial in the control session; mean ± SEM) for each treatment condition ($n = 11$ for each). The data underlying this figure can be found in https://doi.org/10.5281/zenodo.10141750. 5-HIAA, 5-hydroxyindoleacetic acid; CSF, cerebrospinal fluid; DA, dopamine; ITI, inter-trial interval; pCPA, para-chlorophenylalanine.

released a bar when a visual target changed from red to green. A visual cue at the beginning of each trial indicated the amount of reward they could get (1, 2, 4, or 8 drops). All monkeys had been trained to perform basic color discrimination trials on a cued multi-trial reward-schedule

task [17] for more than 3 months. As in previous experiments using a single option task, the required action was very easy, and monkeys could not fail if they actually tried to release the bar at the proper time (the error rate is indeed much lower in the absence of information about costs and benefits) [2]. As in previous experiments in which costs and benefits were manipulated, errors (either releasing the bar too early or too late) were usually observed in small reward trials and/or close to the end of daily sessions [2,18,19]. Note that error trials were repeated with the same cue-reward condition, which prevented the monkeys from skipping unwanted trials. Therefore, bar "errors" were considered to have occurred when the monkeys were not sufficiently motivated to release the bar at a time that would lead to reward. The frequency of error trials is thus a reliable metric for quantifying the influence of motivation on behavior [15,18,19]. Furthermore, we have previously shown that the error rate ($E$) is inversely related to the reward size ($R$), which has been formulated with a single free parameter $a$ [2] (Fig 1F),

$$E = 1/aR. \tag{1}$$

This inverse relationship was consistently observed in the control condition in all monkeys (e.g., CON in Fig 1D and 1E). After pCPA treatment, error rates increased independently from reward-size-related errors. For example, in monkey BO, the error rate became progressively higher in sessions that followed the first and second treatments, while differences that depended on reward size appeared to remain the same (Fig 1D, pCPA-day 1 and 2). A reward-independent increase in error rate was consistently found in all monkeys tested, as shown in the average plot of the normalized error rate (Fig 1E). We next quantified how much of the increase in errors was related to reduced incentive (i.e., devalued reward) and how much was reward-size independent. These factors can be captured by a decrease in parameter $a$ (reward impact or incentive) of the inverse function and implementation of the intercept $e$, respectively (Fig 1F and 1H). To quantify the increase in error rate, we compared 5 models that considered these 2 factors as random effects: Model #1, random effect on $a$; Model #2, random effect on a fixed $e$; Model #3, independent random effects on both $a$ and $e$; Model #4, a single normal distribution of random effect on $a$ and $e$; and Model #5, random effect on $e$ (see S3 Table). Model #4 was selected as the best model with the lowest Bayesian information criterion (BIC) value (S3 Table), indicating that the increase in error rate was explained by simultaneous changes in parameters $a$ and $e$. Our model-based analysis revealed that, compared with the pre-treatment baseline, $a$ was significantly lower on day 2 of pCPA treatment (one-way ANOVA, main effect of treatment, $F_{(2, 20)} = 4.6$, $p = 0.023$; post hoc Tukey HSD, $p = 0.025$ for pCPA-day2 versus CON; Fig 1G). At the same time, parameter $e$ was significantly higher on pCPA-day 2 (main effect of treatment, $F_{(2, 20)} = 4.1$, $p = 0.031$; post hoc Tukey HSD, $p = 0.025$ for pCPA-day 2 versus CON; Fig 1I). These results suggest that the 5-HT depletion-induced increase in errors can be explained by 2 components: one is reduced incentive ($a$), and the other is a factor that appears orthogonal to the incentive value (increase in parameter $e$). We hypothesized that the value-independent component reflected expected cost and subsequent tests supported this interpretation (see Effects of 5-HTR blockade on cost-based motivation below).

Given that reward value (and thus incentive) decreases as animals become satiated [19], we further investigated how the error rate increased along with satiation. In Fig 2A, the average error rates from the normalized data ($n = 11$) are replotted as a function of normalized cumulative reward (see Materials and methods). As previously shown, overall error rates in the control condition increased for each reward size as the normalized cumulative reward increased. This satiation-dependent change in error rate was commonly observed among the 3 conditions, with the effect being stronger in the 2 posttreatment sessions (pCPA-day1 and -day2)

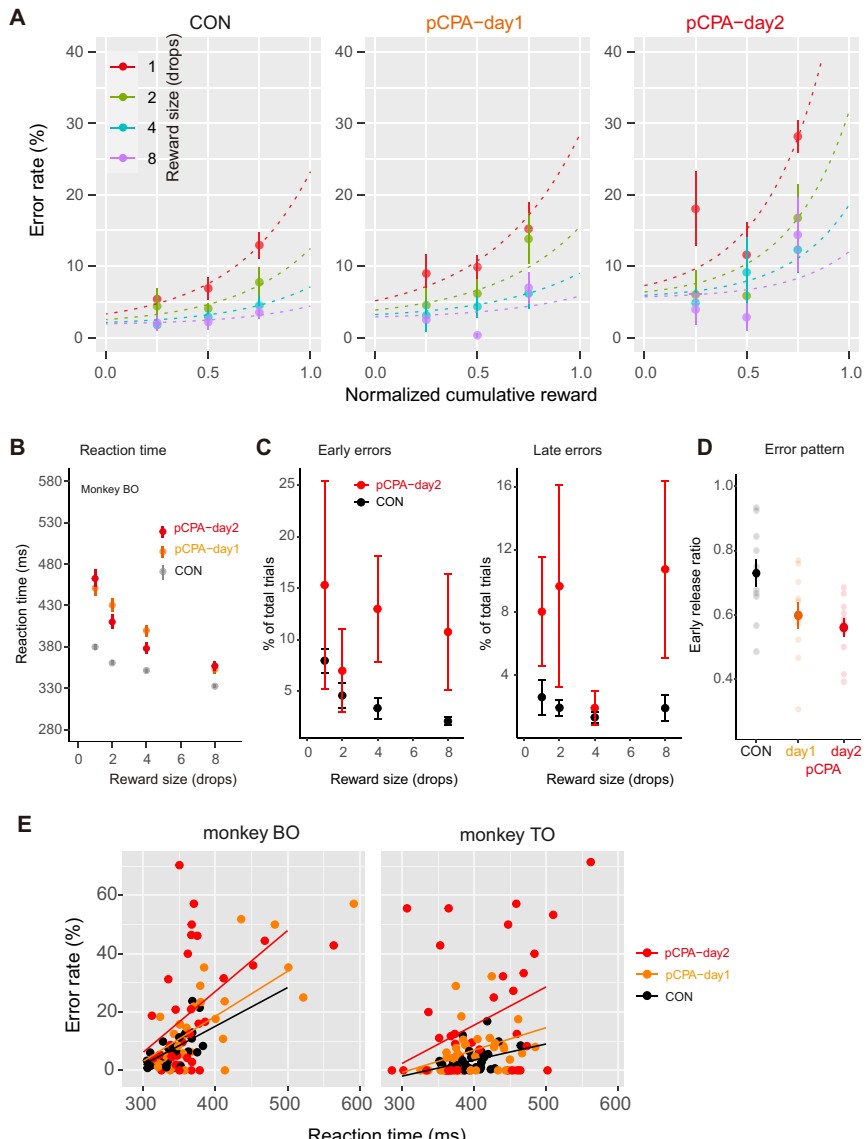

**Fig 2. Effect of 5-HT depletion on impact of satiation on error, RT, error pattern, and RT-error relationship.** (A) Error rate (mean ± SEM; $n = 11$ sessions) as a function of normalized cumulative reward for control and pCPA treatment conditions. Colors indicate reward size. Curves indicate the best-fit model of Eq 4. (B) Mean RT (mean ± SEM) as a function of reward size for control (CON) and pCPA treatment conditions in monkey BO. Colors indicate treatment condition. (C) Rate of early and late errors (mean ± SEM) for control and pCPA-day2 conditions. (D) Early release rate (mean ± SEM) for control and pCPA treatment conditions. (E) Relationship between error rate and mean RT for each reward size in the first and second half of each session under pCPA treatment in monkeys BO and TO, respectively. Colors indicate treatment condition. Colored lines represent the best-fitting linear regression models that explained the data (S4 Table). The data underlying this figure can be found in https://doi.org/10.5281/zenodo.10141750. pCPA, para-chlorophenylalanine; RT, reaction time.

and corresponding to reduced incentive due to 5-HT depletion (*cf.*, reduced *a*, in Fig 1G). However, we also observed pronounced increases in posttreatment error rates that were independent of satiation level; namely, error rates became higher even during the early phase of the session, presumably when thirst drives were still high (Fig 2A). Indeed, fitting the data to an error-rate model that incorporates the satiation effect (Eq 4) showed that regardless of reward size or satiation level, the error rates were higher in posttreatment sessions (*e* = 3.3 and 6.6 for

pCPA-day1 and -day2, respectively) than in the control session ($e = 0.74$). This again suggests that 5-HT depletion blunts motivation in a manner partially independent from reduced incentive.

We also examined changes in reaction time (RT) across trials as another behavioral measure of motivation. Consistent with previous studies [20,21], 5-HT depletion prolonged RTs (Fig 2B). Indeed, RTs were significantly longer with repeated injections [two-way ANOVA, main effect of treatment, $F_{(2, 110)} = 7.1$, $p < 0.01$; post hoc Tukey's honest significant difference (HSD), $p < 0.01$ for day1 versus CON and day2 versus CON] without significantly interacting with reward size (main effect of reward size, $F_{(3, 110)} = 6.0$, $p < 0.001$ interaction, $F_{(6, 110)} = 0.40$, $p = 0.88$). In addition to the slower responses, pCPA treatment also significantly decreased general motor activity in the home cage (4 sessions in 3 monkeys; one-way ANOVA, main effect of treatment, $F_{(1,11)} = 7.28$, $p = 0.021$). Administering pCPA increased both early and late errors across all reward-size conditions (Fig 2C), with late errors tending to increase more (e.g., Fig 2D; one-way ANOVA, main effect of treatment, $F_{(2, 20)} = 11.9$, $p < 0.001$; post hoc Tukey HSD, $p < 0.005$ for day1 versus CON and day2 versus CON). A simple explanation for these effects is that 5-HT depletion affects motivation, which is a common source of control over all behaviors related to performing the task. Thus, whether monkeys perform the action correctly (error rate) and how quickly they respond (RT) should be similarly affected. We reasoned that, if this were the case, the intersession variability in RT and error rate should be correlated and consistent across conditions. However, a session-by-session analysis revealed that the changes in these 2 behavioral measures did not follow the prediction; although there was a linear relationship between error rates and RTs in each treatment condition, the slopes of the linear relationship became steeper as pCPA treatment was repeated (Fig 2E and S4 Table). This finding suggests that additional factors beyond the normal motivational processes at work in goal-directed behavior may contribute to increased error.

## Effects of 5-HTR blockade on incentive

Next, we sought to identify the receptor subtype(s) contributing to the incentive-dependent and incentive-independent (putative cost-dependent) decrease in motivation. We performed PET imaging with selective radioligands for 5-HT$_{1A}$ ([$^{11}$C]WAY100635), 5-HT$_{1B}$ ([$^{11}$C]AZ10419369), 5-HT$_{2A}$ ([$^{18}$F]altanserin), and 5-HT$_4$ ([$^{11}$C]SB207145), and quantified specific radioligand binding using a simplified reference tissue model [22] with the cerebellum as the reference region. Consistent with previous human studies [16], different patterns of receptor distribution were observed for each subtype. For example, high levels of 5-HT$_{1A}$R expression were observed in the anterior cingulate cortex, amygdala, and hippocampus, lateral prefrontal cortex, and posterior cingulate cortex, while its expression was relatively sparce in the basal ganglia (Fig 3A). High 5-HT$_{1B}$R expression was observed in the occipital cortex, ventral pallidum, and substantia nigra (Fig 3B). High 5-HT$_{2A}$R expression was observed in the mediodorsal prefrontal and occipital cortex, while subcortical expression was minimal (Fig 3C). Binding of 5-HT$_4$ was mainly observed in the striatum (Fig 3D).

Selective antagonists are available for these 4 receptor subtypes, each of which is pharmacologically capable of attenuating or blocking 5-HT transmission. Because pharmacological profiles such as receptor affinity and bioavailability differ, the same level of inhibition cannot be achieved with identical dosages of these antagonists. Thus, to compare the effect of receptor blockade between subtypes, we first determined the appropriate dosages. We previously used receptor occupancy as an objective measure for this purpose [15]. To mimic a reduction in 5-HT transmission comparable to the serotonin depletion observed in the CSF after pCPA treatment (approximately 30%), we measured tracer binding at baseline and after systemic

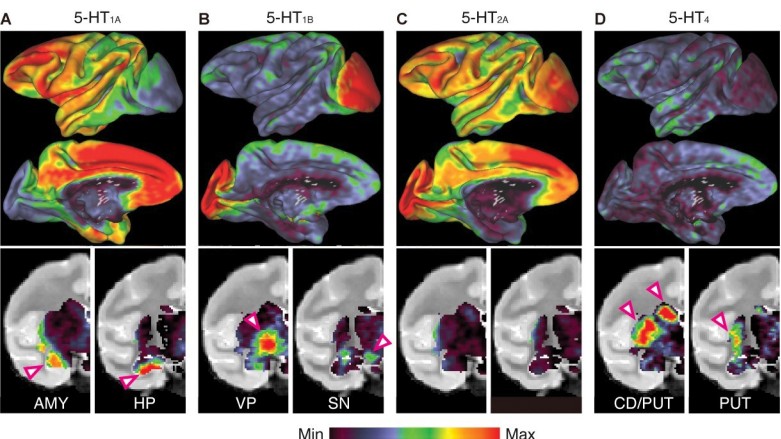

**Fig 3. Heterogeneous distribution of 5-HTR subtypes as measured by PET.** Top: Average density maps for four 5-HTR subtypes (A–D; 5-HT$_{1A}$R, 5-HT$_{1B}$R, 5-HT$_{2A}$R, and 5-HT$_4$R, respectively) on the inflated common macaque surface. Lateral (top) and medial (bottom) views of the left hemisphere are shown. Bottom: Subcortical distribution of the four 5-HTRs shown on 2 representative coronal slices. Color scale represents 2nd–98th percentile of receptor density (i.e., radiotracer binding potential). AMY, amygdala; HP, hippocampus; VP, ventral pallidum; SN, substantia nigra; CD, caudate nucleus; PUT, putamen; PET, positron emission tomography. The data underlying this figure can be found in https://doi.org/10.5281/zenodo.10141750.

administration in 3 monkeys (S1 Table and S1 Fig) and established the dose of antagonist required to achieve approximately 30% to 40% blockade (receptor occupancy), except for 5-HT$_{2A}$, which achieved over 50% (Table 1).

We next examined the effects of separately blocking the receptor subtypes on performance in the reward-size task in 3 monkeys. We administered 3 of 4 different 5-HTR antagonists or vehicle as a control (S2 Table) to each monkey. Blockade of 5-HT$_{1A}$R substantially increased the error rate (Fig 4A), while blockade of 5-HT$_{1B}$R resulted in a moderate increase (Fig 4B). Model-based analysis revealed that blockade of 5-HT$_{1A}$R significantly affected both incentive (*a*) and the reward-size independent factor (putative cost; *e*) (Model #29 in S5 Table)—reducing *a* to 40% and increasing *e* by about 17% on average (Fig 4E and 4F). Blockade of 5-HT$_{1B}$R significantly reduced *a* to 70% on average (Fig 4E; Model #30, S5 Table). In contrast, blockade of 5-HT$_{2A}$R did not significantly affect either (Fig 4C; Model #25, S5 Table). This was not because the blockade level was too low, as another high dose of the antagonist (achieving approximately 84% occupancy) did not change the error rates (S2 Fig). Blocking 5-HT$_4$R consistently decreased error rates (Fig 4D) and significantly increased *a* to approximately 150% (Fig 4E; Model #30, S5 Table). Thus, it is unlikely that 5-HT$_{2A}$R or 5-HT$_4$R directly mediates the behavioral changes observed with 5-HT depletion. Together, these results suggest that the incentive-dependent and incentive-independent components of the lower motivation (as quantified by increased error rate) that we observed upon 5-HT depletion were reproduced by

**Table 1. Antagonist dosage and occupancy for each 5-HT receptor subtype.**

| Receptor | PET ligand | Antagonist | Dose (mg/kg) | Occupancy |
|---|---|---|---|---|
| 5-HT$_{1A}$ | [$^{11}$C]WAY100635 | WAY100635 | 0.3 | 37% |
| 5-HT$_{1B}$ | [$^{11}$C]AZ10419369 | GR55562 | 1 | 34% |
| 5-HT$_{2A}$ | [$^{18}$F]altanserin | MDL100907 | 0.002 | 54% |
| 5-HT$_4$ | [$^{11}$C]SB207145 | GR125487 | 1 | 32% |

The data underlying this table can be found in https://doi.org/10.5281/zenodo.10141750.

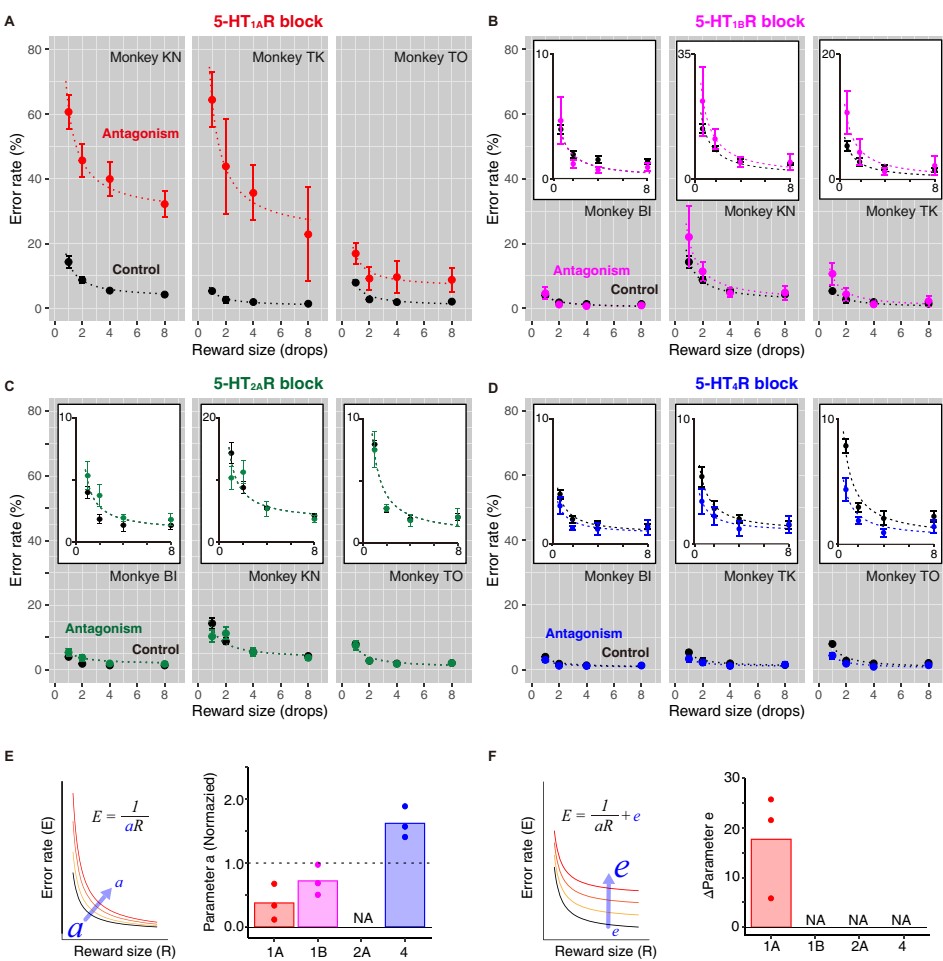

**Fig 4. Effects of 5-HTR blockade on incentive.** (A–D) Error rates (mean ± SEM) as a function of reward size in 3 monkeys after blocking 5-HT$_{1A}$R, 5-HT$_{1B}$R, 5-HT$_{2A}$R, and 5-HT$_4$R by systemic administration of specific antagonists (colored points) and saline control (black points) in 3 monkeys, respectively (S2 Table). Colored and black dashed curves indicate the best fit inverse functions summarized in S5 Table. Rescaled plots are shown in insets in B–D. (E, F) Summary of parameter changes for *a* (incentive) and *e* (value-independent error), respectively. Dots and bars indicate individual data and mean values, respectively. Dashed line at 1 indicates the value for the control. NA denotes that the parameter was not selected as a random effect in the best-fit model (S5 Table). The data underlying this figure can be found in https://doi.org/10.5281/zenodo.10141750.

blocking two 5-HTRs: incentive (reward impact, *a*) was reduced by blocking 5-HT$_{1A}$R or 5-HT$_{1B}$R, whereas a factor orthogonal to the incentive value (increased parameter *e*, putative cost) was increased exclusively by blocking 5-HT$_{1A}$R.

Blockade of 5-HT$_{1A}$R prolonged RTs independent of reward size (two-way ANOVA, main effect of treatment, $F_{(1, 214)} = 366$, $p < 10^{-16}$; treatment × reward size, $F_{(1, 3)} = 1.1$, $p = 0.33$; S3A Fig). In contrast, RTs were not altered by blocking any of the other 5-HTRs ($p > 0.05$; S3B–S3D Fig). The error pattern did not change with any of the treatments (S4 Fig). A session-by-session analysis revealed that the linear relationship between error rates and RTs was altered exclusively by 5-HT$_{1A}$R blockade (S5 Fig and S6 Table), again reproducing the behavioral changes observed after 5-HT depletion (Fig 2D). Thus, altering 5-HTR transmission via 5-HT$_{1A}$R appeared to mediate the component of the motivational processes that was unrelated to reduced incentive, and which had been observed with 5-HT depletion, accurately mimicking changes in the error rate–incentive function and error rate–RT relationship.

## Effects of 5-HTR blockade on cost-based motivation

Our results so far suggest that a reduction in 5-HT transmission via 5-HT$_{1A}$R leads to a decrease in motivation that is partially independent of incentive (reward size/satiation). However, we have not yet established what this factor is or what it depends on. As motivation is the integration of incentive for rewards and the expected cost required to obtain them, and because motivation decreases with higher expected costs, we hypothesized that the emergence of the incentive-independent factor (parameter $e$ in our model) represents an overestimation of expected cost. To test this hypothesis, we directly investigated the effect of selective 5-HTR blockade on cost-based motivation using a work/delay task (Fig 5A) that had the same basic features as the reward-size task, but instead of changing reward amounts, manipulated 2 types of expected costs separately on each trial [18]. In "work" trials, the monkeys had to perform 0, 1, or 2 additional instrumental trials (for details, see Materials and methods) to obtain a fixed amount of reward. In "delay" trials, after the monkeys correctly performed an instrumental trial, a reward was delivered after a 0- to 7-s delay. The number of extra trials or length of the delay was indicated by a visual cue presented throughout the trial. In the first trial after the reward, the visual cue indicated how much would have to be paid to get the next reward. Therefore, we assessed performance on the first trials to evaluate the impact of expected cost on motivation and decision-making. We have previously shown that monkeys exhibit linear relationships between error rate ($E$) and remaining cost ($CU$) for both work and delay trials, as follows:

$$E = kCU + e, \tag{2}$$

where $k$ is a coefficient and $e$ is an intercept [18] (Fig 5B). Extending the inference and formulation of the reward-size task (Eq 1), this linear effect proposes that the reward value is hyperbolically discounted by the cost, with coefficient $k$ corresponding to the discounting factors.

We tested 3 monkeys (S1 and S2 Tables) and measured their error rates to infer delay and workload discounting. We confirmed that the error rates in the control condition increased as the remaining costs increased (e.g., Fig 5C, control). Fig 5B illustrates 2 possible mechanisms by which 5-HTR blockade might increase error rates. First, it could increase cost sensitivity (i.e., discounting factor, $k$; expected cost), which appears as an increase in error rate relative to the remaining cost (Fig 5B, left). Alternatively, errors might increase irrespective of the cost (i.e., increased $e$; Fig 5B, right). As when we analyzed effects on incentive, we compared the effect of 5-HTR antagonism on task performance when receptors were blocked to the same degree (i.e., approximately 30% occupancy; Table 1). Linear mixed model (LMM) analysis tested the assumption that 5-HTR blockade independently increased delay and workload discounting without considering the random effect of treatment condition (e.g., Figs 5C and S6 and S7 Table; see Materials and methods). We extracted the discounting factors and summarize their changes due to blockade in Fig 5D. Blockade of 5-HT$_{1A}$R significantly increased delay discounting in 2 of 3 monkeys and workload discounting in all 3 cases, whereas it significantly increased cost-independent error in only 1 monkey (Figs 5D and S6B). In contrast, blockade of 5-HT$_{1B}$R significantly decreased delay discounting in only 1 of 3 cases (S6B Fig). Blockade of either 5-HT$_{2A}$R or 5-HT$_4$R did not significantly impact the discounting factors.

In the control condition, RT was longer as the remaining cost increased. Blockade of 5-HT$_{1A}$R significantly enhanced the effect of expected cost on RT regardless of cost type (three-way ANOVA, main effect of treatment, $F_{(1, 358)} = 92.2$, $p < 10^{-16}$; treatment × remaining cost, $F_{(1, 2)} = 6.1$, $p = 0.002$; treatment × trial type, $F_{(1, 1)} = 3.6$, $p = 0.057$; S7 Fig) in all but one case (monkey ST). In contrast, blocking the other subtypes did not prolong RT ($p > 0.05$; S7 Fig). None of the treatments changed the error pattern ($p > 0.05$;

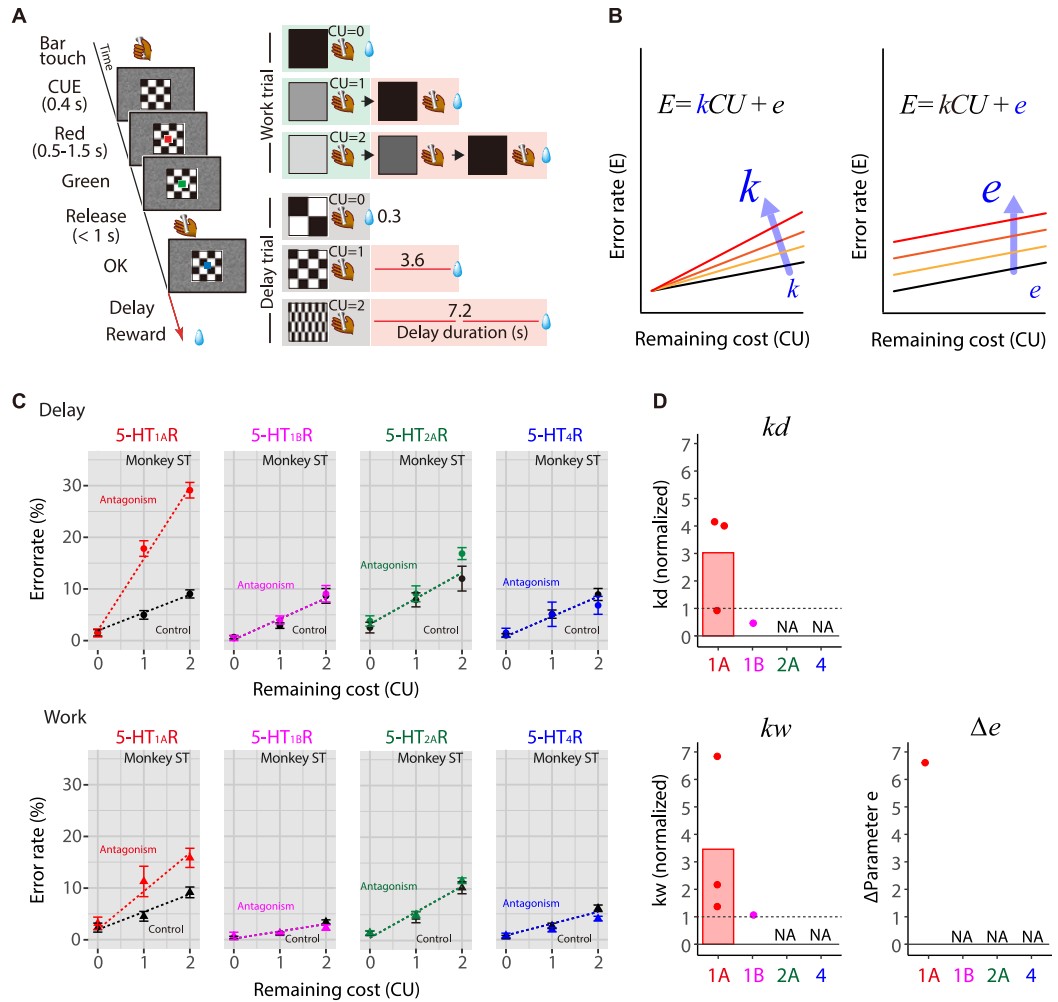

**Fig 5. Effect of 5-HTR blockade on cost-based motivation.** (A) The work/delay task. Left: The sequence of events. A monkey initiated a trial by touching the bar in the chair. After 100 ms, a visual cue representing "cost required" was presented at the center of the monitor. After 500 ms, a red target also appeared at the center of the monitor. After a variable interval of 500–1,500 ms, the target turned green, indicating that the monkey could release the bar to receive a reward. If the monkey responded between 200 and 1,000 ms, the target turned blue indicating the trial had been completed correctly. On correct trials, 1 drop of water reward was delivered immediately or after a delay (3.6 or 7.2 s). An ITI of 1 s was enforced before the next trial could begin. If the monkey made an error by releasing the bar before the green target appeared, within 200 ms after it appeared, or failed to respond within 1 s, all visual stimuli disappeared, the trial was terminated immediately, and the trial was repeated after the 1-s ITI. Right: Relationships between visual cues and trial timing in work trials (top 3 rows on the right) and delay duration in delay trials (bottom 3 rows on the left). Trials outlined by green boxes were used for the analysis. CU denotes the remaining (arbitrary) cost unit necessary to obtain a reward, i.e., either the remaining trials to perform (workload) or the remaining delay periods to wait through. (B) Schematic of model for increases in error rate related to increased cost sensitivity ($k$) or increased sensitivity to baseline cost ($e$). (C) Representative relationships between error rates (monkey ST; mean ± SEM) and remaining costs for delay (top) and work trials (bottom). Data for saline control (black line), 5-HT$_{1A}$R, 5-HT$_{1B}$R, 5-HT$_{2A}$R, and 5-HT$_4$R blockade (colored lines) are shown from left to right, respectively. Colored and black dashed lines indicate the best-fit linear models described in S7 Table. (D) Comparison of the effects of 5-HTR blockade on the workload-discounting parameter ($k_w$), delay-discounting parameter ($k_d$), and sensitivity to baseline cost ($e$). Dashed line at 1 indicates the value for the control. NA denotes that the parameter was not selected as a random effect in the best-fit model (S7 Table). The data underlying this figure can be found in https://doi.org/10.5281/zenodo.10141750. ITI, inter-trial interval.

S8 Fig). Thus, increased expected cost can consistently account for the behavioral manifestation of reduced motivation due to 5-HT$_{1A}$R antagonism, i.e., both prolonged RT and increased error rates.

## Discussion

5-HT deficiency has been shown to decrease motivation and alter cost sensitivity in humans. Although studies in rodents have provided insights into how 5-HT signaling impacts various aspects of motivational behavior, a knowledge gap remains between these basic findings and behavioral disorders in humans. Using macaque monkeys, the present study aimed to fill this gap and identify specific roles of 5-HTR subtypes in reducing motivation. We found that reduced 5-HT transmission leads to reduced motivation through 2 distinct processes: increased cost sensitivity exclusively via 5-HT$_{1A}$R and reduced incentive via 5-HT$_{1A}$R and 5-HT$_{1B}$R. Our findings shed light on the mechanistic role of 5-HT in different aspects of motivational regulation and their relevance to different aspects of medications used to treat depression.

### Two factors decrease motivation for goal-directed behavior

Previous behavioral pharmacological studies demonstrated that 5-HT depletion slows action and/or reduces the likelihood of engaging in behavior [20,21,23], suggesting decreased motivation. However, these previous studies did not directly examine the effect of 5-HT depletion on incentive by using multiple incentive conditions. Thus, data describing the quantitative relationships among 5-HT, reward, and motivation are unavailable. In the present study, we formulated and quantified the relationship using a behavioral paradigm (reward-size task), in which the behavioral measure, error rates, reflects the incentive portion of motivation. Using the same paradigm, we previously demonstrated that DA receptor blockade (either D1- or D2-like receptors) increased the rate of error and was explained by a decrease in incentive (i.e., lower subjective reward, indicated by a smaller parameter *a*) [15]. In the current study, although 5-HT depletion also increased the error rate, the rate was only partially explained by reduced incentive. We also observed a factor orthogonal to the incentive value (increased parameter *e*). In addition, error rates increased constantly regardless of the satiation level. Thus, our results suggest that 5-HT depletion results in both incentive-dependent and incentive-independent reductions in motivation. Subsequent experiments using another behavioral paradigm (work/delay task) revealed that the incentive-independent factor is likely the cost.

Importantly, our pharmacological studies replicated and disentangled incentive-dependent and cost-dependent factors; the former was observed exclusively after 5-HT$_{1A}$R and 5-HT$_{1B}$R blockade and the latter after 5-HT$_{1A}$R antagonism. These behavioral effects can be directly compared among the 4 major 5-HTR subtypes because we used PET to fix the antagonist occupancy level across subtypes. Thus, our findings suggest 2 independent mechanisms for 5-HT depletion-induced loss of motivation that are mediated by 2 receptor subtypes (5-HT$_{1A}$R and 5-HT$_{1B}$R), which PET indicated are distributed in different brain regions. As discussed below, our results highlight a new view of the 5-HT system at the receptor level with respect to differential aspects of motivational control.

We also emphasize that our results suggest that multiple factors and related brain processes may drive a single behavioral measure of motivation, namely, an increase in error rate. This may be important from a clinical perspective, as there is a tendency to use general terms such as "decreased motivation" or "anhedonia" to subsume all aspects of motivational impairment into 1 broad construct. Our results challenge this oversimplified view of motivational impairment and propose a more nuanced view in which multiple behavioral factors and neural processes underlie it.

### Reduced 5-HT transmission via 5-HT$_{1B}$R leads to decreased motivation in a value-dependent manner

The incentive-dependent component of decreased motivation was reproduced by both 5-HT$_{1A}$R and 5-HT$_{1B}$R blockade, with the latter showing a relatively stronger effect. Blockade

of 5-HT$_{1A}$R produced both value-dependent and -independent decreases in motivation, highlighting a central role in decreased motivation. However, unlike 5-HT$_{1A}$R, 5-HT$_{1B}$R blockade did not affect cost sensitivity. Similar value-dependent reductions in motivation were observed following DA receptor antagonism with either D$_1$R or D$_2$R blockade [15]. Given the mutual interactions between 5-HT and DA systems [24,25], the behavioral changes observed here may be mediated, at least in part, by DA. 5-HT$_{1B}$R is abundant in the basal ganglia, particularly in the output structures (globus pallidus and substantia nigra) (Fig 3B). Moreover, activation of 5-HT$_{1B}$R increases DA release in the mesocorticolimbic system in rats [26,27]. Blockade of 5-HT$_{1B}$R attenuates cocaine-seeking, but not natural food-intake [28], suggesting that 5-HT transmission via 5-H$_{1B}$R is related to outcome-based motivational control.

Primate visual cortex is rich in 5-HT$_{1B}$R [29], which has been implicated in contrast-based edge/contour detection [30], a potential mediator of the observed behavioral changes. However, the use of high-contrast stimuli (i.e., white/black cues and red/green signals) may prevent potential effects due to contrast sensitivity. In fact, the behavioral effects of 5-HT$_{1B}$R blockade on the current study were reward-size dependent, which is contrary to what would happen if it was related to changes in the visual system; analyzing data by reward size would result in uniform effects, not reward-size-dependent ones.

Our findings extend the current view of 5-HT$_{1B}$R function, derived primarily from rodent studies, to primates and emphasize that its contribution to motivation is reward-related rather than cost-related [31]. This has important implications for understanding how 5-HT regulates motivational behavior and its relationship to the pathophysiology and pharmacotherapy of depression and other disorders. A previous study suggested that up-regulation of postsynaptic 5-HT$_{1B}$R in the nucleus accumbens and ventral pallidum may be involved in the antidepressant effect of ketamine [32]. Alteration of 5-HT$_{1B}$R binding has also been reported in some psychiatric disorders, such as post-traumatic stress disorder (PTSD), alcohol dependence, pathological gambling, and drug abuse [33–37]. It is thus important to conduct future studies focusing on the relationship between altered 5-HT$_{1B}$R function and reward sensitivity in humans.

## Reduced 5-HT transmission via 5-HT$_{1A}$R leads to decreased motivation due to overestimation of future costs

One of the key findings of this study was that blockade of 5-HT$_{1A}$R resulted in lower motivation by altering both incentive and non-incentive-related factors. Although errors can occur through a variety of mechanisms such as impulsivity and decreased attention, our results suggest that the incentive-independent changes after blockade may be related to an overestimation of the impending cost; when we manipulated costs (workload or a delay before reward) independently of incentive value, the impact of cost on motivation increased following 5-HT$_{1A}$R blockade (Fig 5). These observations led us to conclude that reduced 5-HT transmission via 5-HT$_{1A}$R results in increased cost sensitivity and decreased incentive, both of which act to reduce motivation.

5-HT has been implicated in the control of temporal discounting, which is the part of the cost-benefit evaluation process that regulates motivation. In humans, a decrease in 5-HT caused by tryptophan depletion has been shown to increase the rate of delayed reward discounting [9,38]. Conversely, 5-HT reuptake inhibitors (SSRIs, presumably by up-regulating 5-HT) decrease delay discounting [39]. Similarly, 5-HT depletion induced by pCPA administration tended to cause rats to choose immediate small rewards over long-delayed large rewards [7], whereas administration of an SSRI was associated with a decrease in impulsive choices [6]. In addition to 5-HT depletion, reduced 5-HT transmission mediated by 5-HT$_{1A}$R

results in increased cost sensitivity to delayed reward; specifically, administration of a 5-HT$_{1A}$R antagonist (WAY100635) tended to cause rats to choose immediate small rewards over long-delayed large rewards [6]. In monkeys, 5-HT$_{1A}$R blockade decreased correct responses as the delay to reward increased [12]. However, other studies suggest mixed results, with relatively weak or no effect of 5-HT$_{1A}$R antagonism on delayed discounting [40–42]. This may be because those studies confounded incentive and cost factors in each option.

In contrast to delay sensitivity, the involvement of 5-HT in valuation of other types of costs, such as effort, is less clear or controversial. A rodent study of cost-benefit trade-offs showed that SSRIs reduced effort expenditure [43]. Another study also reported similar effects of SSRIs, including suppression of higher effort activities such as lever pressing and wheel running [44]. Other rodent studies have shown that 5-HT depletion with pCPA had no effect on the tendency of rats to make cost-based choices [7,45]. One human study showed that patients taking SSRIs produced more effort for monetary incentives, which was mediated by a reduction in the cost of effort, but no change in the weights of incentives [5]. In contrast, several human studies have reported that SSRIs are relatively ineffective in treating motivational dysfunctions such as fatigue and anergia [44,46–50].

As our PET data show (Fig 3), 5-HT$_{1A}$Rs are predominantly expressed in the limbic system, such as the medial prefrontal cortex (mPFC), amygdala, and hippocampus, which is in good agreement with human data [16]. The binding potential of 5-HT$_{1A}$R, including in these regions, is diminished in patients with major depressive disorder [51,52] and monkeys exhibiting depression-like behavior [53], suggesting that alteration of 5-HT transmission is a pathophysiology associated with depression and probably its behavioral phenotype, including effortful information processing. However, additional studies are needed to identify the brain circuitry contributing to increased cost sensitivity via 5-HT$_{1A}$R.

5-HT$_{1A}$R blockade induced increases in error rate during the reward-size task, with individual variation—the increase was high in 2 monkeys and relatively small in another (Fig 4). Despite the quantitative differences, these effects were qualitatively consistent across individuals in the sense that they could be decomposed into 2 factors: lower incentive and another factor orthogonal to incentive values. Because 1 monkey exhibited relatively weaker effects for both factors, the variation may be due to pharmacological effects, including differences in 5-HT$_{1A}$R expression level and/or antagonist pharmacokinetics.

In the current study, other receptors were found to have limited or no involvement in the increased cost sensitivity. However, several previous rodent studies suggest a contribution of 5-HT$_{2A}$R in the mPFC to impulsivity [54–56]. Another rodent study showed that antagonism of 5-HT$_{2C}$R enhanced effort-based motivation [57], suggesting that it mediates decreases rather than increases in motivation. Similarly, the current study showed that blockade of 5-HT$_4$R significantly increased motivation, suggesting that normal 5-HT transition via 5-HT$_4$R decreases incentive (Fig 4E). Importantly, this implies that 5-HT$_4$R is a potential target for treating loss of motivation. Our results, together with other studies, indicate that different 5-HTR subtypes likely modulate motivation in different directions, suggesting a more complicated view of what SSRIs do in humans, as they are assumed to enhance 5-HT transmission equally at multiple receptor subtypes.

Studies examining the activity of 5-HT neurons and their optogenetic manipulation also encourage us to consider motivation-independent factors beyond cost sensitivity. 5-HT neurons convey information not only about reward but also about aversive events [58,59]. In addition, another example of complexity is the involvement of 5-HT in orchestrating behavioral inhibition [60]. Future investigation will be needed to examine the consequence of manipulating 5-HT transmission in nonhuman primates in the context of reward and punishment.

## Limitations of this study

First, because we administered antagonists systemically, the current study could not determine which brain areas are responsible for antagonist-induced changes in incentive and expected cost. While our findings, particularly the differential localization of receptor subtypes (Fig 3), support the idea that different limbic structures are involved in incentive and cost sensitivity, further research (such as local infusion of 5-HTR antagonists) is needed to identify key loci and determined the circuitry and molecular mechanisms underlying 5-HT's role in motivation.

Second, because our receptor blockade was partial and incomplete, the motivational involvement of other receptor subtypes may have been overlooked. For example, previous studies reported the involvement of $5\text{-HT}_{2A}R$ in temporal discounting; specifically, systemic administration of the mixed $5\text{-HT}_{2A}R$ agonist DOI dose-dependently impaired the ability to wait, whereas the $5\text{-HT}_{2A}R$ antagonist ketanserin blocked the impulsivity effect of DOI [61]. In the present study, relatively high blockade of $5\text{-HT}_{2A}R$ (approximately 50% and 80% occupancy) did not alter incentive motivation or cost sensitivity. However, this does not rule out the possible involvement of $5\text{-HT}_{2A}R$ in these functions, as has been previously suggested.

Third, it should be noted that the present study was conducted only in male monkeys. Given that sex differences in 5-HT function are known [62], future studies should examine whether the present results are applicable to female monkeys.

Finally, because $5\text{-HT}_{1A}R$ and $5\text{-HT}_{1B}R$ are located in the cell soma and presynaptic terminals of 5-HT neurons and serve as autoreceptors that regulate 5-HT neuronal activity and release [63–65], systemically administering their antagonists may affect serotonin transmission in multiple ways, not only reducing postsynaptic receptor activation but also increasing overall 5-HT signaling. Despite this complexity, behavioral changes resulting from blockade of these receptors partially mimicked those observed after 5-HT depletion, suggesting that the reduction in motivation was mainly due to decreased 5-HT signaling via $5\text{-HT}_{1A}R$ and $5\text{-HT}_{1B}R$. Clear dissection of presynaptic regulatory mechanisms, especially in primate models, is challenging with currently available technology. Therefore, further investigation using advanced/innovative methods will provide deeper insights into the intricate interplay between 5-HT signaling and receptor subtypes.

## Conclusion

The present study demonstrates a differential contribution of receptor subtypes to reduced motivation caused by decreased 5-HT transmission: $5\text{-HT}_{1A}R$ mediates increased cost sensitivity and decreased incentive, whereas $5\text{-HT}_{1B}R$ only mediates reduced incentive. Because these 2 receptor subtypes are differentially distributed in limbic brain areas, their differential contribution might indicate that separate 5-HT circuitry underlies distinct aspects of the valuation processes that regulate motivation for action. Taken together, our findings increase our understanding of how 5-HT signaling impacts motivation in terms of cost-benefit trade-offs, thus providing an advanced framework for understanding the pathophysiology and medications used to treat psychiatric disorders.

## Materials and methods

### Ethics statement

All experimental procedures involving animals were conducted in accordance with the Guide for the Care and Use of Nonhuman Primates in Neuroscience Research (The Japan Neuroscience Society; https://www.jnss.org/en/animal_primates) and approved by the

Animal Ethics Committee of the National Institutes for Quantum Science and Technology (#09–1035).

## Subjects

A total of 15 adult male macaque monkeys (13 Rhesus and 2 Japanese; 4.5 to 7.9 kg; aged 4 to 12 years at the start of the experiment; see S1 Table for a summary of subjects) were used in this study. Food was available ad libitum, and motivation was controlled by restricting access to fluid to experimental sessions in which water was provided as a reward for task performance. Animals received water supplementation whenever necessary (e.g., when they were unable to obtain sufficient water through experimentation) and had free access to water whenever testing was interrupted for more than 1 week. For environmental enrichment, play objects and/or small foods (fruits, nuts, and vegetables) were provided daily in the home cages.

## Drug treatment

For 5-HT depletion, monkeys were injected intraperitoneally with pCPA solution (C3635, Sigma-Aldrich, 0.9% in saline) at a dose of 150 mg/kg/day for 2 consecutive days. We examined the effect of 5-HT depletion on behavior by analyzing the behavioral parameters in the 2 consecutive days before pCPA treatment as control (CON) and those in the days following the first and second treatments (pCPA-day1 and -day2, respectively; Fig 1A). Although we did not perform a control test in the same cohort, we looked at the saline injection data and confirmed that the act of injection itself did not affect error rates in the 2 days after injection [two-way ANOVA, main effect of days, $F_{(2, 138)} = 2.3$, $p = 0.10$]. Total motor activity in the home cage was also analyzed after the behavioral test until room lights were turned off each day (16:00 to 21:00 h) after the control and post-pCPA sessions. Motor activity was recorded using motion-sensitive accelerometers (ActiCal, Motion Biosensors, PHILIPS) attached to the monkeys' collars.

For 5-HTR blockade, the following 5-HTR antagonists were used: WAY100635 (W108, Sigma-Aldrich; for 5-HT$_{1A}$R), GR55562 (cat# 1054, Tocris; for 5-HT$_{1B}$R), MDL100907 (M3324, Sigma-Aldrich; for 5-HT$_{2A}$R), and GR125487 (cat# 1658, Tocris; for 5-HT$_4$R). WAY100635, GR55562, and GR125487 were dissolved in 0.9% saline, while MDL100907 was suspended in a drop of hydrochloric acid and the final volume was adjusted with saline. The dose for each antagonist is listed in Table 1. Monkeys were pretreated with one of the antagonists intramuscularly 15 min before the start of the behavioral test or PET scan. For behavioral testing, saline was injected as a control using the same procedure. The volume administered was set at 1 mL for all experiments.

5-HTR blockade was performed 4 times per individual antagonist in the behavioral test. Each vehicle or antagonist was administered once per week, with days of the week counterbalanced. After completing an antagonist test, the monkeys' baseline task performances and conditions were assessed, including daily activity in the home cage, body weight, and water and food consumption. If there were no abnormalities, the next sequence of behavioral tests was initiated with a different antagonist. The order of treatment of the 4 antagonists was counterbalanced for each monkey.

## Measuring 5-HT metabolites and DA in the CSF

Acute CSF samples were collected by lumbar puncture with a 23-gauge needle under ketamine–xylazine anesthesia. CSF was collected twice (0.5 μl/sample per monkey): once a week before pCPA administration and again the day after 2 consecutive days of pCPA

administration. The supernatant was passed through a 0.22-μm filter (Millex-GV, Merck, Germany), centrifuged at 2,190 × g for 20 min, and aliquoted into another microcentrifuge tube. For analysis of 5-HIAA and DA, high-performance liquid chromatography (HTEC-500, EICOM Co., Japan) was used with a monoamine separation column (SC-5ODS, EICOM). CSF data were recorded and analyzed using Power Chrom software (version 2.5, eDAQ, United States of America). The mobile phase was a mixture of 0.1 M phosphate buffer (pH 6.0) and methanol at a ratio of 83:17, including EDTA-2Na (5 mg/L). The amount of monoamine was calculated quantitatively as an absolute amount relative to the monoamine standard solution (MA11-STD, EICOM).

## PET procedure and occupancy measurement

Four monkeys were used for PET measurements, which were performed with 4 PET ligands: [$^{11}$C]WAY100635 (for 5-HT$_{1A}$R), [$^{11}$C]AZ10419369 (for 5-HT$_{1B}$R), [$^{18}$F]altanserin (for 5-HT$_{2A}$R), and [$^{11}$C]SB207145 (for 5-HT$_4$R). PET scans were performed using an SHR-7700 PET scanner (Hamamatsu Photonics, Japan) or microPET Focus220 scanner (Siemens Medical Solutions USA) on conscious monkeys that were seated in a chair. The injected amount of radioactivity and its molar activity at the time of injection were between 88.4 to 360.4 MBq and 14.8 to 296.3 GBq/μmol, respectively. After transmission scans for correcting attenuation, a dynamic emission scan was performed for 90 min, except for scans with [$^{11}$C]SB207145 (120 min). Ligands were injected as a single bolus via the crural vein at the start of the scan. All emission data were reconstructed by means of filtered back projection using a Colsher or Hanning filter. Radioactive concentrations in tissue were obtained from volumes of interest (VOIs) placed on the global cortex, visual cortex, and cerebellum (as a reference region). Each VOI was defined on individual structural magnetic resonance (MR) images (EXCELART/VG Pianissimo at 1.0 Tesla, Toshiba, Japan) that were co-registered with PET images using PMOD image analysis software (PMOD Technologies, Switzerland). The regional radioactivity of each VOI was calculated for each frame and plotted against time. Regional binding potentials relative to non-displaceable radioligands (BP$_{ND}$) of 5-HTRs were estimated using a simplified reference tissue model. Monkeys were scanned with and without drug treatment on different days.

Parametric images of the BP$_{ND}$ were constructed using the original multilinear reference tissue model [66]. Individual structural MR images were registered to the Yerkes19 macaque template [67,68] using FMRIB's linear registration tool (FLIRT) and FMRIB's nonlinear registration tool (FNIRT) implemented in FSL software (FMRIB's Software Library, http://www.fmrib.ox.ac.uk/fsl) [69]. BP$_{ND}$ images were then normalized to the template using structural MRI-to-template matrices. Average BP maps across monkeys were plotted on the surface map [70] using the Connectome Workbench [71]. Because BP$_{ND}$ is assumed to linearly reflect the spatial distribution and availability of receptors throughout the brain region, we rescaled each BP map by its minimum and maximum values to allow comparison of relative receptor distribution.

Occupancy levels were determined from the degree of BP$_{ND}$ reduction by antagonists [72]. 5-HT receptor occupancy was estimated as follows:

$$Occupancy(\%) = (1 - BP_{NDTreatment}/BP_{NDBaseline}) \times 100, \tag{3}$$

where BP$_{ND\ Baseline}$ and BP$_{ND\ Treatment}$ are BP$_{ND}$ measured without (baseline) and with an antagonist, respectively. The target VOI was the visual cortex for 5-HT$_{1B}$R and the global cortex for other subtypes.

## Behavioral tasks and testing procedures

A total of 8 monkeys were used for the behavioral study (see S1 Table). For all behavioral training and testing, each monkey sat in a primate chair inside a sound-attenuated dark room. Visual stimuli were presented on a computer monitor in front of the monkey. Behavioral control and data acquisition were performed using a QNX-based Real-time Experimentation data acquisition system (REX; Laboratory of Sensorimotor Research, National Eye Institute) and commercially available software (Presentation, Neurobehavioral Systems). We used 2 types of behavioral tasks, reward-size and work/delay, as previously described [2,18]. Before the behavioral experiments, all monkeys were trained to perform color discrimination trials on a cued multi-trial reward schedule task [17] for more than 3 months.

In the reward-size task, a monkey initiated a trial by touching the bar in the chair; 100 ms later, a visual cue (13˚ on a side), which will be described below, was presented at the center of the monitor. After 500 ms, a red target (0.5˚ on a side) also appeared at the center of the monitor. After a variable interval of 500, 750, 1,000, 1,250, or 1,500 ms, the target turned green, indicating that the monkey could release the bar to earn a liquid reward. If the monkey responded within 200 to 1,000 ms, the target turned blue, indicating that the trial had been completed correctly. In correct trials, a reward of 1, 2, 4, or 8 drops of water (1 drop = ~0.1 ml) was delivered immediately after the blue signal. Each reward size was selected randomly with equal probability. The visual cue presented at the beginning of the trial indicated the number of reward drops that the monkey would receive (Fig 2A). An inter-trial interval (ITI) of 1 s was enforced before the next trial began.

In the work/delay task, the basic task consisted of a series of color discrimination trials that was the same as in the reward-size task (see Fig 5A). The main difference was that, unlike the reward-size task, we manipulated task requirements but fixed the amount of reward. There were 2 types of trial: work trials and delay trials. In work trials, the monkeys had to perform 0, 1, or 2 additional instrumental trials to obtain the reward, and a visual cue indicated how many trials remained until a water reward (approximately 0.25 ml) was delivered. In delay trials, the monkeys performed 1 color discrimination and a water reward (approximately 0.25 ml) was delivered immediately after each correct signal or after a delay period. The visual cue indicated both the trial type and cost to obtain a reward (Fig 5A). Pattern cues indicated delay trials and the timing of reward delivery after a correct performance: immediately (0.3 s, 0.2 to 0.4 s; mean, range), after a short delay (3.6 s, 3.0 to 4.2 s), or after a long delay (7.2 s, 6.0 to 8.4 s). Grayscale cues indicated work trials and the number of extra trials the monkey would have to perform to obtain a reward. We set delay durations to be equivalent to the duration of 1 or 2 trials of color discrimination so that we could directly compare the cost of 1 or 2 arbitrary units (cost unit; CU) [18]. All other aspects of the task were the same as those in the reward-size task.

In both tasks, if the monkey released the bar before the green target appeared, within 200 ms after it appeared, or failed to respond within 1 s, we regarded the trial as an "error trial." At this point, all visual stimuli disappeared, the trial was terminated immediately, and after a 1-s ITI, the trial was repeated with the same cue-reward/cost condition. Our behavioral measurement of motivation was the proportion of error trials. Before each testing session, monkeys were subjected to approximately 22 h of water restriction in their home cage. Each session continued until the monkey would no longer initiate a new trial (usually less than 100 min). The monkeys were tested with the work/delay task for 1 or 2 daily sessions as training to become familiar with the cueing condition. Each monkey was tested from Monday to Friday. Treatment with 5-HTR antagonist or saline (as a control) was performed 1 day per week.

## Behavioral data analysis

All data and statistical analyses were performed using the R statistical computing environment. The average error rate for each trial type was calculated for each daily session, with error rates for each trial type being defined as the number of error trials divided by the total number of trials of that given type. We did not distinguish between the 2 types of errors (early or late release) and used their sum except for the error pattern analysis. We performed repeated-measures ANOVAs to determine the effect of treatment × reward size (for data in the reward-size task) or treatment × cost type × remaining cost (for data in the work/delay task) on RT and error pattern. Post hoc comparisons were performed using Tukey's HSD test, and a priori statistical significance was set at $p = 0.05$.

We used error rates to estimate the level of motivation, because the error rates of these tasks ($E$) are inversely related to the value for action [2]. In the reward-size task, we used the inverse function (Eq 1). We fit the data to LMMs [73] in which the random effects across 5-HT depletion conditions (i.e., cCPA-day1 and -day2) on parameter $a$ and/or intercept $e$ (Fig 1) were nested. Model selection was based on the BIC, an estimator of in-sample prediction error for nested models (S3 Table). Using the selected model, parameter $a$ and intercept $e$ were estimated simultaneously within each subject. Parameter $a$ was then normalized to the value in the nontreated condition (control, CON). LMMs were also applied for correlation analysis between error rate and RT. For the 5-HT depletion study, mean error rates and RTs were calculated for each reward size in the first and second halves of each session to increase the number of data sets (Fig 2E). Four statistical models were nested to account for the presence or absence of random effects of monkey and treatment conditions and the best-fit model was selected based on BIC (S4 Table).

To examine the effects of satiation, each session was divided into 3 parts based on normalized cumulative reward, $R_{cum}$. Mean error rates in the reward size task across 11 sessions were then fit to the following model:

$$E = \frac{1}{aR \times F(R_{cum})} + e, \tag{4}$$

where the satiation effect, $F(R_{cum})$, as the reward value was exponentially decaying in $R_{cum}$ at a constant $\lambda$ [18]:

$$F(R_{cum}) = e^{-\lambda R_{cum}}. \tag{5}$$

To estimate the effect of 5-HTR blockade on the parameters, we also used LMMs. Models were nested to account for the presence or absence of random effects, random effects of treatment conditions, and subjects (see S5 Table). The best model was selected based on the BIC for the entire dataset, which is the sum of the results for the regression of each unit nested by individual and/or treatment condition. We confirmed the robustness of the parameter(s) by performing the Bayesian inference analysis, which indicates how reliable 2 parameters are in fitting the data (see S9 Fig).

In the work/delay task, we used linear models to estimate the effect of remaining cost, i.e., workloads and delay, as previously described [15,19]:

$$E_w = k_w CU + e, \tag{6}$$

$$E_d = k_d CU + e, \tag{7}$$

where $E_w$ and $E_d$ are the error rates, and $k_w$ and $k_d$ are cost factors for work and delay trials, respectively. $CU$ is the number of remaining cost units, and $e$ is the intercept. We

simultaneously fitted a pair of these linear models to the data by sum-of-squares minimization without weighting. We also used LMMs to estimate the effect of 5-HTR blockade on discounting parameters. We imposed the constraint that the intercept ($e$) has the same value between cost types and assumed a basic statistical model in which the random effects of blocking condition independently affect the regression coefficients. Other methods were the same as those used for data in the reward-size task. Models and results are reported in S7 Table.

## Supporting information

**S1 Table. Summary of subjects used in this study.** A tick indicates the monkey used the experiment. Species (J, Japanese; R, Resus); Sex (F, Female; M, Male); BW, body weight; CSF, cerebrospinal fluid; RS task, reward-size task; W/D task, work/delay task.
(DOCX)

**S2 Table. Summary of drug treatment for each monkey.** Numbers indicate the number of treatments.
(DOCX)

**S3 Table. Model comparison for the effect of 5-HT depletion on error rate (for Fig 1E).** $a$($cond$) and $e$($cond$) indicate the random effects of conditions with 5-HT depletion on parameters $a$ and $e$, respectively. The random effect of Model #3 was assumed to be normally distributed on $a$ and $e$, independently. The random effect of Model #4 was assumed to be a 2D normal distribution on $a$ and $e$: ($1/a_{cond}$, $e_{cond}$) ~ biNorm(0, $\Sigma_{cond}$). BIC is a relative measure of quality for the models (#1–5). ΔBIC denotes the difference from the minimum BIC.
(DOCX)

**S4 Table. Model comparison for the effect of 5-HT depletion on the relationship between error rate and RT in the reward-size task (for Fig 2D).** $a$($mk$), $a$($cond$), $e$($mk$), and $e$($cond$) indicate the random effects of 5-HT depletion conditions on parameters $a$ and $e$, respectively. The probability distribution of the random effects were as follows: $a_{mk}$ and $a_{cond}$ were ~N(0, $\sigma^2_{mk}$) and ~N(0, $\sigma^2_{cond}$), respectively. ($a'_{mk}$, $e'_{mk}$) and ($a'_{cond}$, $e'_{cond}$) were ~biNorm(0, $\Sigma_{mk}$) and ~biNorm(0, $\Sigma_{cond}$), respectively. $E$, error rate; $RT$, reaction time; $mk$, monkey; $cond$, treatment condition (pCPA-day1, pCPA-day2, or control). BIC is a relative measure of quality for the models (#1–4). ΔBIC denotes the difference from the minimum BIC.
(DOCX)

**S5 Table. Model comparison for the effect of 5-HTR blockade on error rate in the reward-size task (for Fig 4).** $a$($cond$) $a$($mk$), $a$($cond\_mk$), $e$($cond$), $e$($mk$), and $e$($cond\_mk$) indicate the random effects of blocking 5-HTR on parameters $a$ and $e$, respectively. The probability distribution of the random effects were as follows: $a_{cond}$, $a_{mk}$, and $a_{cond\_mk}$ were ~N(0, $\sigma^2_{cond}$), ~N(0, $\sigma^2_{mk}$), and ~N(0, $\sigma^2_{cond\_mk}$), respectively. ($a'_{cond}$, $e'_{cond}$), ($a'_{mk}$, $e'_{mk}$), and ($a'_{cond\_mk}$, $e'_{cond\_mk}$) were ~biNorm(0, $\Sigma_{cond}$), ~biNorm(0, $\Sigma_{mk}$), and ~biNorm(0, $\Sigma_{cond\_mk}$), respectively. $E$, error rate; $cond$, treatment condition (antagonist or control); $mk$, monkey; $cond\_mk$, interaction of treatment condition and monkey. BIC is a relative measure of quality for the models (#1–30). ΔBIC denotes the difference from the minimum BIC.
(DOCX)

**S6 Table. Model comparison for the effect of 5-HTR blockade on the relationship between error rate and RT in the reward-size task (for S4 Fig).** $a$($mk$), $a$($cond$), $e$($mk$), and $e$($cond$) indicate the random effects of 5-HT depletion on parameters $a$ and $e$, respectively. The probability distribution of the random effects were as follows: $a_{cond}$ and $a_{mk}$ were ~N(0, $\sigma^2_{cond}$) and ~N(0, $\sigma^2_{mk}$), respectively. ($a'_{cond}$, $e'_{cond}$) and ($a'_{mk}$, $e'_{mk}$) were ~biNorm(0, $\Sigma_{cond}$) and ~biNorm

$(0, \Sigma_{mk})$, respectively. $E$, error rate; $RT$, reaction time; $mk$, monkey; $cond$, treatment condition (antagonist or control). BIC is a relative measure of quality for the models. $\Delta$BIC denotes the difference from the minimum BIC.
(DOCX)

**S7 Table. Model comparison for the effect of 5-HTR blockade on error rate in the work/ delay task (for Fig 5C).** $CU$ and $E_0$ indicate the remaining cost and the intercept, respectively. $k(treat)$, $k(type)$, $e(treat)$, and $e(type)$ denote the random effects of blocking 5-HT on parameters $k$ and $e$, respectively. The probability distribution of the random effects were as follows: $k_{treat}$ and $k_{type}$ were $\sim N(0,\sigma^2_{treat})$ and $\sim N(0,\sigma^2_{type})$, respectively. $(k'_{treat}, e'_{treat})$ and $(k'_{type}, e'_{type})$ were $\sim$biNorm$(0, \Sigma_{treat})$ and $\sim$biNorm$(0, \Sigma_{type})$, respectively. $E$, error rate; $treat$, treatment condition (antagonist or control): $type$, trial type (delay or work). BIC is a relative measure of quality for the models (#1–25). $\Delta$BIC denotes the difference from the minimum BIC.
(DOCX)

**S1 Fig. Occupancy estimation.** Example of occupancy estimation based on reduction in specific tracer binding. (A) Representative coronal section from an MR image showing the region of interest for binding measurements (ROI, drawn by purple line). (B, C) Representative coronal sections from parametric PET images showing the specific binding ($BP_{ND}$) of $[^{11}C]$ WAY100635 in monkey CH during baseline (B) and blocking (C) conditions (pretreated with cold WAY100635 at a dose of 0.3 mg/kg, i.m.). Occupancy was determined as the proportion of reduced specific binding relative to baseline [i.e., $(BP_{ND}^{baseline} - BP_{ND}^{blocking})/BP_{ND}^{baseline}$]. In this case, the reduction of specific binding was 36.7%.
(EPS)

**S2 Fig. No discernible effect of high-dose 5-HT$_{2A}$R blockade on error rates in the reward-size task.** Error rates (mean ± SEM) as a function of reward size for control (black) and 5-HTR blockade at high 5-HT$_{2A}$R occupancy (84%) (MDL100907 at 0.03 mg/kg, i.m.; dark green). There was no significant difference in error rates between CON and 5-HT$_{2A}$R blockade (two-way ANOVA, main effect of treatment, $F_{(1, 72)} = 0.54$, $P = 0.46$). Dotted curves indicate the best fit of inverse functions (Model #25 in S5 Table). Data obtained from monkey KN. The data underlying this figure can be found in https://doi.org/10.5281/zenodo.10141750.
(EPS)

**S3 Fig. Effect of 5-HTR blockade on reaction time in the reward-size task.** Mean reaction time as a function of reward size for control (black) and 5-HTR blockade conditions (color). (A) 5-HT$_{1A}$ blockade, (B) 5-HT$_{1B}$R blockade, (C) 5-HT$_{2A}$R blockade, and (D) 5-HT$_{4}$R blockade. The data underlying this figure can be found in https://doi.org/10.5281/zenodo.10141750.
(EPS)

**S4 Fig. Effect of 5-HTR blockade on error pattern in the reward-size task.** Early release rate (mean ± SEM) as a function of reward size for control (black) and 5-HTR blockade conditions (color). (A) 5-HT$_{1A}$ blockade, (B) 5-HT$_{1B}$R blockade, (C) 5-HT$_{2A}$R blockade, and (D) 5-HT$_{4}$R blockade. The data underlying this figure can be found in https://doi.org/10.5281/zenodo.10141750.
(EPS)

**S5 Fig. Effect of 5-HTR blockade on the relationship between error rate and reaction time in the reward-size task.** Session-by-session relationships between error rate and average reaction time for each reward size during control and after 5-HTR blockade for each monkey. Colors indicate treatment condition. Lines represent the best-fit linear regression models that explain the data. The data underlying this figure can be found in https://doi.org/10.5281/

zenodo.10141750.
(EPS)

**S6 Fig. Effect of 5-HTR blockade on reaction time in the work/delay task.** Relationships between error rates (mean ± SEM) and remaining costs for delay (top) and work trials (bottom) for monkeys KY (A) and MP (B), respectively. Data for saline control (black line), $5\text{-HT}_{1A}R$, $5\text{-HT}_{1B}R$, $5\text{-HT}_{2A}R$, and $5\text{-HT}_4R$ blockade (colored lines) are shown from left to right, respectively. Colored and black dashed lines indicate the best-fit linear models described in S7 Table. The data underlying this figure can be found in https://doi.org/10.5281/zenodo. 10141750.
(EPS)

**S7 Fig. Effect of 5-HTR blockade on reaction time in the work/delay task.** Mean reaction time as a function of reward size for control (black) and 5-HTR blockade conditions (color) in delay and work trials. The data underlying this figure can be found in https://doi.org/10.5281/ zenodo.10141750.
(EPS)

**S8 Fig. Effect of 5-HTR blockade on error pattern in the work/delay task.** Late release rate (mean ± SEM) as a function of reward size for control (black) and 5-HTR blockade conditions (color). The data underlying this figure can be found in https://doi.org/10.5281/zenodo. 10141750.
(EPS)

**S9 Fig. Parameter estimation of the data for $5\text{-HT}_{1A}R$ blockade and control in monkey KN.** The red and green dots represent the posterior distribution of the model parameters from an Hamiltonian Monte Carlo simulation (iter_warmup = 2000, iter_sampling = 4000, and chains = 4) for $5\text{-HT}_{1A}R$ blockade and control conditions, respectively. The red and green circles indicate the 95th percentile for each condition. In both conditions, the parameter obtained from the best-fit LMM model (+) are very close to the maximum a posteriori (MAP) estimate from the simulation (black dots), suggesting a high reliability of the parameter estimates. All R-hat values were below 1.02, indicating that this Bayesian simulation was statistically converged. All random effects were statistically significant as the 95% confidence intervals deviated from 0. The data underlying this figure can be found in https://doi.org/10.5281/zenodo. 10141750.
(EPS)

## Acknowledgments

We thank R. Suma, T. Sugii, R. Yamaguchi, Y. Matsuda, and J. Kamei for their technical assistance. We also thank S. Bouret for his comments on the earlier version of the manuscript. The 2 Japanese monkeys used in this study were provided by National Bio-Resource Project "Japanese Monkeys" of MEXT, Japan.

## Author Contributions

**Conceptualization:** Yukiko Hori, Takafumi Minamimoto.

**Formal analysis:** Yukiko Hori, Koki Mimura, Yuji Nagai, Yuki Hori, Takafumi Minamimoto.

**Funding acquisition:** Yukiko Hori, Takafumi Minamimoto.

**Investigation:** Yukiko Hori, Yuji Nagai.

**Resources:** Katsushi Kumata, Ming-Rong Zhang.

**Writing – original draft:** Yukiko Hori, Takafumi Minamimoto.

**Writing – review & editing:** Yukiko Hori, Koki Mimura, Yuji Nagai, Yuki Hori, Tetsuya Suhara, Makoto Higuchi, Takafumi Minamimoto.

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
