## [Editor Report · Decision Letter 0]

6 Mar 2023

Dear Dr Minamimoto, 

Thank you for submitting your manuscript entitled "Reduced serotonergic transmission alters sensitivity to cost and reward via 5-HT1A and 5-HT1B receptors in monkeys" for consideration as a Research Article by PLOS Biology.

Your manuscript has now been evaluated by the PLOS Biology editorial staff, as well as by an academic editor with relevant expertise, and I'm writing to let you know that we would like to send your submission out for external peer review.

Once your full submission is complete, your paper will undergo a series of checks in preparation for peer review. After your manuscript has passed the checks it will be sent out for review. To provide the metadata for your submission, please Login to Editorial Manager (https://www.editorialmanager.com/pbiology) within two working days, i.e. by Mar 08 2023 11:59PM.

Kind regards,

Roli Roberts

Roland Roberts, PhD

Senior Editor

PLOS Biology

rroberts@plos.org

---

## [Decision Letter · Decision Letter 1]

21 May 2023

Dear Takafumi,

Thank you for your patience while your manuscript "Reduced serotonergic transmission alters sensitivity to cost and reward via 5-HT1A and 5-HT1B receptors in monkeys" was peer-reviewed at PLOS Biology. It has now been evaluated by the PLOS Biology editors, an Academic Editor with relevant expertise, and by three independent reviewers. Please accept my apologies for the time that this has taken.

In light of the reviews, which you will find at the end of this email, we would like to invite you to revise the work to thoroughly address the reviewers' reports.

You'll see that reviewer #1 is broadly positive, but identifies “some critical concerns and limitations” – among their hierarchically arranged “major points,” it seems that some would need further analyses, and others involve methodological clarifications and more comprehensive stats. Reviewer #2 is very positive, and only has presentational requests. Reviewer #3 finds the study “impressive,” but has a number of concerns about the framing and interpretation. Their concerns seem to centre around problems with 5HT1B blockade and model-fitting (s/he makes some helpful-sounding recommendations).

Given the extent of revision needed, we cannot make a decision about publication until we have seen the revised manuscript and your response to the reviewers' comments. Your revised manuscript is likely to be sent for further evaluation by all or a subset of the reviewers.

**IMPORTANT - SUBMITTING YOUR REVISION**

*Re-submission Checklist*

*Published Peer Review*

*PLOS Data Policy*

*Blot and Gel Data Policy*

Sincerely,

Roli

Roland Roberts, PhD

Senior Editor

PLOS Biology

rroberts@plos.org

REVIEWERS' COMMENTS:

Reviewer #1:

The serotonin (5-HT) system is believed to be a significant factor in depression and other psychiatric disorders that involve decreased motivation. However, the precise role of 5-HT in motivation remains unclear. In this study, researchers manipulated the 5-HT system in monkeys using drugs and gauged the effects on motivation in a goal-directed task. They discovered that inhibiting 5-HT synthesis reduced motivation, which could be separated into value-dependent and value-independent components. The researchers also employed specific drugs to block different 5-HT receptor subtypes and found that blocking 5-HT1A receptors reduced motivation in a value-independent manner, whereas blocking only 5-HT1B receptors lessened the impact of incentive value on motivation. These results imply that the 5-HT system has a complex role in motivation, with various receptor subtypes mediating distinct processes that lead to reduced motivation in 5-HT deficiency. Although the experimental procedure is commendable, there are some critical concerns and limitations. We think that the authors can address these after revising the manuscript. 

Major points:

(1) How to relate "refusal rate" with motivation. 

(1-1) The paper's definition of motivation appears to be ambiguous, and its interpretation of cost, reward, and motivation is the main weakness. The authors used "refusal rate" as a measure of motivation, which is closely linked to the Go-Nogo movement process and may not fully capture the complexity of motivational processing. 

(1-2) Moreover, the current definition of "Refusal rate" encompasses various types of errors, including early release before the green target appeared and late release after the green target. It would be beneficial for the authors to separate these errors based on their occurrence before or after the green target and present the results separately. These errors may have different underlying meanings, with early releases possibly indicating impulsivity and late releases possibly indicating decreased attention. Additionally, the monkeys' release of the bar to indicate their cost evaluation may be prone to errors, such as releasing the bar even if they want to continue the trial or not releasing the bar if they are not paying enough attention to the task. 

(1-3) While the mathematical modeling of the refusal rate is of high quality and meaningful, it is challenging to fully credit it as a measure of motivational processing. Therefore, it is essential for the authors to clearly dissociate the various factors contributing to refusal rate, including types of movement (i.e., go-nogo, preparation and/or initiation of movement), cost, motivation, and error. If the authors cannot separate these factors, they should at least acknowledge their limitations in the discussion section.

(2) The experimental results in this paper require clearer descriptions.

(2-1) In Fig. 1E, it would be helpful to include information about the number of sessions per animal. It appears that the authors used data from 11 sessions for each condition (control, day 1, and day 2), obtained from 4 monkeys. Knowing which monkey contributed to each session would be beneficial.

(2-2) Regarding Fig. 4, it is noteworthy that 5HT4R also increased parameter "a." The authors could provide their opinion on this result to help readers understand its significance. Additionally, statistical analysis could be performed for Fig. E and F to support the findings.

(2-3) In Fig. 5D, it would be useful to perform statistical analysis to demonstrate the significance of 5HT1A.

(2-4) Finally, in the introduction section, it would be beneficial to explain more about why incentives and costs are important in motivation and 5HT systems. Currently, these concepts appear suddenly and may be difficult to understand in relation to the 5HT systems.

Minor points

1. Did the administration of pCPA have any effects on the monkeys' behavior outside of task performance, such as changes in their behavior in their cages? If so, it would be helpful to provide some examples of these observations in the text.

2. Regarding Fig. 3, in addition to the distribution of 5HT receptor subtypes in the brain, could you also provide information about their total amount in the brain and body?

3. On page 12, line 273, it is mentioned "Therefore, blockade of 5HT1BR". Is this supposed to be "5HT1AR" instead?

4. The study used male monkeys exclusively. Could you discuss whether this could potentially affect the results and their interpretation?

5. In Figure 4A, there appears to be a significant variation among the monkeys. Could you discuss this observation in more detail in the discussion section?

6. Could you consider the possibility that 5HT1B occupancy in the occipital cortex may have affected the results, potentially influencing the visual system and causing illusions? This could be discussed in the limitations or future directions section of the paper.

Reviewer #2:

This was an interesting manuscript, which reported on the role of specific 5-HT receptor subtypes in aspects of motivation. The experiments were well designed and well conducted. My specific comments, which are intended to help the authors with the presentation of their results, are listed below.

Major Comments:

The authors have a short paragraph in the discussion mentioning some of the rodent studies focusing on the effects of SSRIs on effort based choice and motivation, as well as one human paper. Although the authors state that "A rodent study" involving SSRIs showed effects on cost based decision making, a more recent paper also showed the same type of effects, including tests of both lever pressing and wheel running (Presby et al. 2021). Moreover, selection of this one human paper is somewhat limited, because the authors do not cite the multiple papers from the clinical literature showing that SSRIs are relatively poor at treating motivational dysfunction in depressed people (Papakostas et al. 2006; Fava et al. 2014; Cooper et al. 2014; Rothschild et al. 2014; Popovic et al. 2015), and can even induce fatigue as an adverse event in some people. Since the introduction and discussion of this manuscript focus extensively on the role of the 5-HT in depression and the effects of SSRIs, especially motivational aspects, the references cited above are relevant for showing the conflicting results about the role of 5-HT in motivational dysfunction in depression. The human literature itself is complicated, so the findings do not seem to be simply understood in terms of a rodent vs. primate difference.

In the discussion, the authors discuss their results showing that no other 5-HT receptors were involved in effort-related aspects of motivation, and cite one rodent study involving 5-HT 2A receptors. However, Bailey et al. (2018) showed that 5-HT 2C antagonism enhanced effort-based motivation in rats. Interestingly, this points to the involvement of a 5-HT receptor in mediating decreases in motivation rather than increases. Thus, it is possible that different 5-HT receptor subtypes are pushing motivational function in different directions. This finding also suggests a more complicated view of what SSRIs are doing in humans, since they enhance 5-HT transmission at multiple receptor subtypes via blockade of 5-HT transport. 

The results demonstrated dissociations between different aspects of motivation. This should be a central focus of the discussion, because there is a tendency to use general terms such as "anhedonia" to subsume all aspects of motivational impairments into one broad construct. The present pattern of results emphasizes the dissociable nature of different aspects of motivation. 

Reviewer #3:

In this article, the authors aim to dissect the contributions of different serotonin receptor subtypes to processing to cost and rewards. This is an important topic and a well-executed study. There is extensive evidence that serotonin plays an important role in motivation, yet the majority of studies have not addressed receptor specificity, which is particularly important in the case of serotonin, given the large nr of different receptors (relative to e.g. the dopamine system). This has potential clinical implications also given that reduced motivation for effortful tasks may arise through different cognitive and neurobiological mechanisms (e.g. in apathy, chronic fatigue, depression, all associated with serotonin). The authors completed a very impressive set of studies, combining global serotonin depletion (pCPA), PET localization of the different receptor subtypes, and drug administration, across a set of 2 tasks. 

That said, I have a number of concerns regarding primarily the framing and interpretation of the results, as well as a few more technical questions regarding the computational modelling. 

The primary finding as the authors present this, is a double dissociation of the effects of 5HT1A vs 5HT1B blockade, where the former appears to affect the evaluation of costs, while the latter mediates the impact of incentive value on motivation. However, when evaluating the results, I find the effects of 5HT1A antagonism very convincing, but the effects of 5H1B blockade (and interpretation there-of) much less so (figure 4, figure 6). This is for a number of reasons: 

- The 1B effects are much smaller, and appear to be drive by purely the lowest reward sizes. This may be because the base refusal rate (figure 4) is very low already (varies a little bit per monkey, but for most it has asymptoted by a reward size of 2-4. Thus, if 1B blockade lowers refusal rates, it only has a very small 'window' in which it can do this. 

- This dissociation hinges on the model comparison where you separately fit a and e. How reliably can both a and e can be fitted, given the low refusal base rate and early asymptote (i.e. at low reward sizes already)? Please include parameter recovery analyses to show that these can indeed be recovered independently in the parameter regimes that you report here (see also Collins & Wilson, 2019 eLife 10 simple rules for computational modelling), particularly with the floor effects of the refusal rates. 

- Furthermore, for the computational modelling model comparison, the authors use the AIC, and find evidence that the model with most parameters is also the winning model. Generally, AIC penalizes very 'lightly' for model complexity, thereby favouring more complex models. If you use the somewhat more conservative BIC, does the same model win? 

Even when the effects of 1B turn out to be indeed robust, the authors should highlight the vast difference in effect size, and tone down the double dissociation nature of the conclusions, but rather focus on the 1A effects. 

MINOR POINTS

Please include colour legends in all figures. 

In the figures with reward size, please also include the '0': I assume the lowest reward size is 1? 

The pCPA study shows increasing refusal rates over the course of testing. Did you also collect a control sample that just completed 3 testing sessions, as this effect may simply be a function of time / nr of sessions completd, rather than pCPA. If no control group was included, please make sure to highlight this caveat. 

Figure 1A: describe the task in more detail in the caption. 

Figure 1C: the term 'early release rate' from the caption is much more informative than 'error rate' - I would recommend putting that in the figure also. 

Figure 5: which monkeys are these plots referring to? All only in monkey ST? 

Can you add a discussion as to what you believe the effects of the different drugs, particularly the 5HT1A and 1B antagonis, have on overall serotonin signalling? 

It is difficult to work out exactly how many datasets were collected for each part of the study. Please add to the main manuscript an overview table to show which manipulations / measures were collected in which monkeys. 

Drug doses were targeted to achieve 30-40% receptor occupancy. Please include the rational for these occupancy levels. You do mention that you aim to achieve the same level of occupancy per drug, but does this mean that these are also the required doses to observe an effect? Could perhaps the absence of effects in some of the drugs (esp. 5HT4) be the result of a too-low occupancy (i.e. underdosing)?

Please explain the tasks in more detail in the main manuscript, rather than referring to previous publications. both in the figure captions showing the task as well as the methods section. Please provide more detail as to how effortful this task is to the monkey, i.e. how strongly this task is tapping into effort processing. 

The serotonin literature cited is relatively old (most papers at ~ 20 years) and primarily focused on the delay discounting / waiting literature. I suggest to also link to the more recent theorizing on the role of serotonin in aversive inhibition and aversive 'pruning' by e.g. Quentin Huys, and relevant rodent optogenetics work by Cohen and colleagues on punishment and reward processing.

---

## [Decision Letter · Decision Letter 2]

3 Nov 2023

Dear Dr Minamimoto,

Thank you for your patience while we considered your revised manuscript "Reduced serotonergic transmission alters sensitivity to cost and reward via 5-HT1A and 5-HT1B receptors in monkeys" for consideration as a Research Article at PLOS Biology. Your revised study has now been evaluated by the PLOS Biology editors, the Academic Editor, and two of the original reviewers.

In light of the reviews, which you will find at the end of this email, we are pleased to offer you the opportunity to address the remaining points from reviewer #3 in a revision that we anticipate should not take you very long. We will then assess your revised manuscript and your response to the reviewers' comments with our Academic Editor aiming to avoid further rounds of peer-review, although might need to consult with the reviewers, depending on the nature of the revisions.

IMPORTANT - please also attend to the following:

a) Please mention the study species in Abstract (i.e. macaques, rather than just "monkeys").

b) Many thanks for providing the data underlying the Figure in your GitHub deposition (http://github.com/minamimoto-lab/2023-Yukiko-5HTR). However, because Github depositions can be readily changed or deleted, please make a permanent DOI’d copy (e.g. in Zenodo) and provide this URL (see below).

c) Please cite the location of the data clearly in all relevant main and supplementary Figure legends, e.g. “The data underlying this Figure can be found in S1 Data” or “The data underlying this Figure can be found in https://doi.org/10.5281/zenodo.XXXXX”

d) Please make any custom code available, either as a supplementary file or as part of your Zenodo deposition.

e) Please remove your funding information from the manuscript file and paste it into the submission form.

**IMPORTANT - SUBMITTING YOUR REVISION**

*Resubmission Checklist*

*Published Peer Review*

*PLOS Data Policy*

*Blot and Gel Data Policy*

Sincerely,

Roli Roberts

Roland Roberts, PhD

Senior Editor

PLOS Biology

rroberts@plos.org

REVIEWERS' COMMENTS:

Reviewer #1:

The authors fully address my concerns. 

Reviewer #3:

The authors comprehensively addressed most of my comments and clearly put in a lot of work. There are a few outstanding issues I would like to follow up on: 

While I appreciate the Bayesian inference analysis (should be included in the paper) of the parameter robustness, I would still like to see parameter recovery. Note that my question here pertains to the model, and thus is independent of the sample size: even for a single set of parameters (i.e. 2 values of a and e) you can simulate e.g. 300 datasets and fit the model to those simulated datasets, and then report the average and spread of the parameter estimates fitted to the simulated data. You can repeat this procedure for each monkey / drug condition. This will give a good qualitative sense of how recoverable the parameters are. 

Model comparison: if I understand correctly, in model 4 the random effects of a and e were coupled, while in model 3 they were independent? This means that if a went up by amount X, then e would go up by the same amount X? It would be good to make this even clearer in the text. Also in talbe S3, the notation of the model is exactly the same: E = 1/a(cond) R + e(cond), so it is unclear what the difference is between these models. 

I am happy to see that the authors have updated their model comparison approach with the BIC. However, I would strongly suggest to report both AIC and BIC, as these offer a sort of 'upper' and 'lower' bound on how much to penalize for model complexity (in reality BIC is often too strict as parameters are rarely fully independent). The difference in model evidence for the models 3 and 4 is only 2.2, which is substantially lower than what is normally considered to be strong evidence in favour of one model over another. This is indeed confirmed by the fact that for the AIC, the preferred model is model 3. In all, this points to the fact that one just cannot draw a strong conclusion in favour of either model 3 or 4. Thus, I would like to see both he AIC and BIC reported in table S3, and the conclusion to be updated that there is just not strong evidence in favour of either model. I do agree with the approach of the authors that the most 'conservative' conclusion is to go with model 4 as there is at least no strong evidence for a double dissociation. 

For figure 5, I don't see a reason to not simply present the data from all monkeys, to give the reader a complete overview of the (variability of) the effects. I would therefore like to see the data for all monkeys that were tested for each condition - these can just be colour (or saturation) -coded and included in the same graph.

---

## [Editor Report · Decision Letter 3]

22 Nov 2023

Dear Dr Minamimoto,

Thank you for the submission of your revised Research Article "Reduced serotonergic transmission alters sensitivity to cost and reward via 5-HT1A and 5-HT1B receptors in monkeys" for publication in PLOS Biology. On behalf of my colleagues and the Academic Editor, Matthew Rushworth, I'm pleased to say that we can in principle accept your manuscript for publication, provided you address any remaining formatting and reporting issues. These will be detailed in an email you should receive within 2-3 business days from our colleagues in the journal operations team; no action is required from you until then. Please note that we will not be able to formally accept your manuscript and schedule it for publication until you have completed any requested changes.

Sincerely, 

Roli Roberts

Senior Editor

PLOS Biology

rroberts@plos.org